# Why Emergent Communication is Repulsive

## Abstract

With the success of deep reinforcement learning, there has been a resurgence of interest in situated emergent communication research. Properties of successful emergent communication have been identified which typically involve auxiliary losses that ensure a trade-off between ensuring diversity of message-action pairs, conditioned on observations, and consistency, when the reward acquired is significant. In this work, we draw theoretically connections between these auxiliary losses and the probabilistic framework of repulsive point processes. We show how in fact these auxiliary losses are promoting repulsive point processes, as well as outline ways in which the practitioner could utilise these repulsive point processes directly. We hope this newfound connection between language and repulsive point processes offers new avenues of research for the situated language researcher or probabilistic modeller.

## 1 Introduction

Deep Reinforcement Learning (DRL) has seen successes in many problems such as video and board games (Mnih et al., 2013; Silver et al., 2016; 2017; Mnih et al., 2015), and control of simulated robots (Ammar et al., 2014; Schulman et al., 2015; 2017). Though successful, these applications assume idealised simulators and require tens of millions of agent-environment interactions, typically performed by randomly exploring policies. However, on the time scales of physical (i.e., real-world) systems, sample-efficiency naturally becomes a more pressing concern due to time and cost burdens.

Sampling in supervised learning is typically used on a fixed dataset $\mathcal{D}$, where mini-batches are sampled from $\mathcal{D}$ to perform parameter updates. The supervised learning literature features a range of *biased* (non-uniform) sampling approaches. Csiba & Richtárik (2018) and Katharopoulos & Fleuret (2018) develop importance sampling schemes that reduce the training time of deep neural networks by orders of magnitude. Zhao & Zhang (2014) motivates the need for *diverse* (in terms of classes) mini-batches and shows that sampling from Repulsive Point Processes (RPP) yields reduced training time and more accurate models. RPPs are probabilistic models on finite sets that can naturally trade-off between quality and diversity when sampling subsets of items. Sampling subsets of items arises in many domains within reinforcement learning, from sampling experience in procedurally generated games to sampling actions and messages in emergent communication.

With the advent of DRL, there has been a resurgence of interest in situated emergent communication (EC) research Das et al. (2017); Lazaridou et al. (2016); Kottur et al. (2017); Jaques et al. (2019); Havrylov & Titov (2017). In this setup, one typically has at least two agents which are de-centralised, yet have a communication channel between them that might be detrimental to the overall performance of both agents, i.e. one agent might be able to see but not move, and needs to guide another agent towards a goal. However, there remain many open design decisions, each of which may significantly bias the nature of the constructed language, and any agent policy which makes use of it. The properties of successful emergent communication were identified in Jaques et al. (2019); Eccles et al. (2019b); Lowe et al. (2019); Cowen-Rivers & Naradowsky (2020), and these typically involve a trade-off between ensuring diversity of message-action pairs, conditioned on observations, and consistency, when the reward acquired, is considerable.

In this work, we discuss the connections between RPPs and emergent communication. We examine properties of successful emergent communication and explain how they in-fact encourage a repulsive point processes over

the actions/ messages. We then show how one could create specifically repulsive emergent communication for either a speaker or listener agent, and detail how this formulation theoretically bias's an agent to speak or listen in a situated language game.

## 2 Why Emergent Communication is Repulsive

First, we introduce the relevant background required. In 2.1 we introduce the background for single agent reinforcement learning, in 2.2 we extend the formulation of reinforcement learning (in 2.1) to emergent communication. We then detail the key elements of diversity and quality. Lastly, in 2.3 we formally introduce Determinantal Point Processes, a computationally efficient repulsive point process, later used to re-define an optimal listener and optimal speaker.

### 2.1 Reinforcement learning

We consider Markov decision processes (MDPs) with continuous states and action spaces; $MDP = \langle \mathcal{O}, \mathcal{A}, \mathcal{P}, c, \gamma \rangle$, where $\mathcal{O} \subseteq \mathbb{R}^{d_{\text{state}}}$ denotes the state space, $\mathcal{A} \subseteq \mathbb{R}^{d_{\text{act}}}$ the action space, $\mathcal{P} : \mathcal{O} \times \mathcal{A} \to \mathcal{O}$ is a transition density function, $c : \mathcal{O} \times \mathcal{A} \to \mathbb{R}$ is the reward function and $\gamma \in [0, 1]$ is a discount factor. At each time step $t = 0, \dots, T$, the agent is in state $\boldsymbol{o}_t \in \mathcal{O}$ and chooses an action $_t \in \mathcal{A}$ transitioning it to a successor state $\boldsymbol{o}_{t+1} \sim \mathcal{P}(\boldsymbol{o}_{t+1}|\boldsymbol{o}_t, _t)$, and yielding a reward $\boldsymbol{r}_t = c(\boldsymbol{o}_t, _t)$. Given a state $\boldsymbol{o}_t$, an action $_t$ is sampled from a policy $\pi : \mathcal{O} \to \mathcal{A}$, where we write $\pi(_t|\boldsymbol{o}_t)$ to represent the conditional density of an action. Upon subsequent interactions, the agent collects a trajectory $\boldsymbol{\tau} = [\boldsymbol{o}_{0:T}, \boldsymbol{a}_{0:T}]$, and aims to determine an optimal policy $\pi^\star$ by maximising total expected reward: $\mathbb{E}_{\boldsymbol{\tau} \sim p_\pi(\boldsymbol{\tau})}[\mathcal{C}(\boldsymbol{\tau})] := \mathbb{E}_{\boldsymbol{\tau} \sim p_\pi(\boldsymbol{\tau})}[\sum_{t=0}^T \gamma^t \boldsymbol{r}_t]$, where $p_\pi(\boldsymbol{\tau})$ denotes the trajectory density defined as: $p_\pi(\boldsymbol{\tau}) = \mu_0(\boldsymbol{o}_0) \prod_{t=0}^{T-1} \mathcal{P}(\boldsymbol{o}_{t+1}|\boldsymbol{o}_t, _t)\pi(_t|\boldsymbol{o}_t)$, with $\mu_0(\cdot)$ being an initial state distribution.

### 2.2 Emergent Communication

A common approach to emergent communication is to concatenate the incoming observational message ($\boldsymbol{o}^m$) and state observation ($\boldsymbol{o}$) together Lowe et al. (2019); Foerster et al. (2016) to create an augmented observational space $\hat{\boldsymbol{o}} = [\boldsymbol{o}, \boldsymbol{o}^m]$. Given a state $\hat{\boldsymbol{o}}_t$, a discrete message $\boldsymbol{m}_t \in \mathcal{M}$ is sampled from a policy $\pi : \mathcal{O} \to \mathcal{M}$, where we write $\pi(\boldsymbol{m}_t|\hat{\boldsymbol{o}}_t)$ to represent the conditional probability distribution of a message given an observation and incoming message. An agent will also have an additional communication (message) policy $\pi(\boldsymbol{m} \mid \boldsymbol{o})$. The replay buffer ($\mathcal{B}$) in emergent communication can be described as a collection of tuples ($\mathcal{X} \in \mathcal{B}$) such that $\mathcal{B} = \{\mathcal{X}_0 := (\boldsymbol{o}_0, \boldsymbol{o}'_0, \boldsymbol{o}_0^m, \boldsymbol{o}'_0{}^m, \boldsymbol{a}_0, \boldsymbol{m}_0, \boldsymbol{r}_0), \dots, \mathcal{X}_n := (\boldsymbol{o}_n, \boldsymbol{o}'_n, \boldsymbol{o}_n^m, \boldsymbol{o}'_n{}^m, \boldsymbol{a}_n, \boldsymbol{m}_n, \boldsymbol{r}_n)\}$.

One of the difficulties in emergent communication is efficiently exploring complex observation, action, communication spaces whilst trading off consistency of actions and messages. There exist auxiliary losses that pressure the speaking/ listening agent to alter its short-term behaviour in response to messages consistently (e.g., causal influence of communication loss Lowe et al. (2019); Jaques et al. (2019); Eccles et al. (2019b)), but bear with them the additional difficulties of tuning extremely sensitive auxiliary loss parameters, as is the case with most auxiliary loss functions. Thus, there is a need for more simplified emergent communication algorithms that achieve success in challenging language games with less sensitivity to hyperparameters. We will now discuss identified characteristics of **Positive Listening** Lowe et al. (2019) and **Positive Signalling** Lowe et al. (2019) that successfully aid communication and how auxiliary loss functions that encourage these losses are in-fact biasing the agents' action distribution towards a repulsive point processes.

**Positive Listening** Its important for an agent to adapt to changes in incoming communication signals/ messages. An agent exhibits positive listening if the probability distribution over its actions is influenced by the messages it receives. The causal influence of communication (CIC) Lowe et al. (2019); Jaques et al. (2019); Eccles et al. (2019b) loss can be defined below

$$\mathcal{CIC} = \mathcal{D}_{KL}(\pi(\boldsymbol{a} \mid \boldsymbol{o}) \mid\mid \pi(\boldsymbol{a} \mid \boldsymbol{o}, \boldsymbol{o}^m)) \tag{1}$$

Where one can marginalise over messages in order to approximate $\pi(\boldsymbol{a} \mid \boldsymbol{o}) = \int_{\boldsymbol{m}} \pi(\boldsymbol{a} \mid \boldsymbol{o}, \boldsymbol{o}^m)$. It is easy to observe, that when a CIC loss is minimised, an agent should have large probability distribution changes when an incoming message is received vs when one is not. One can readily see the repulsive nature of this loss, as the loss biases the agent to taking diverse actions when diverse incoming messages are present in the observation space. Note, this loss consistently encourages diversity, and has no quality term which enables the agent to trade-off between diversity and quality (reward) when the reward received increases, unlike the auxiliary loss for positive listening shown in Eq 2.

**Positive Signalling**    An agent must be consistent with outgoing communication signals. Positive signalling is defined as a positive correlation between the speaker's observation and the corresponding message it sends, i.e., the speaker should produce similar messages when in similar situations. Various methods exist to measure positive signalling, such as speaker consistency, context independence, and instantaneous coordination Lowe et al. (2019); Eccles et al. (2019a). An example of a method for biasing agents towards positive signalling is via the mutual-information ($\mathcal{I}$) loss Eccles et al. (2019a), as shown below. This loss biases the speaker to produce high entropy distribution of overall messages, but when conditioned on the speaker's observation has a low entropy, allowing for exploration and consistency between communication signals.

$$\mathcal{I}(\boldsymbol{m}, \boldsymbol{o}) = \mathcal{H}(\boldsymbol{m}) - \mathcal{H}(\boldsymbol{m}|\boldsymbol{o}) \tag{2}$$

$\mathcal{H}(\boldsymbol{m})$ can be approximated by taking the entropy of the average message distribution and $\mathcal{H}(\boldsymbol{m}|\boldsymbol{o})$ can be computed easily. The repulsive property in this loss is less obvious: we have a diversity promoting term on the left, which promotes a more uniform spread of messages, with a quality term on the right that allows for certain messages, when conditioned on an observation, to maintain a level of consistency (rather than diversity). This loss function therefore trades off between diversity of message-observation pairs, as well as allowing consistency for ones receiving high reward, which a detrimental point process is similarly able to achieve naturally.

## 2.3  Determinantal Point Process

A repulsive point process is a type of probability distribution whereby meaning sampled points are encouraged to repel from previously sampled points with respect to some distance metric. Determinantal Point Process's (DPP's) are a type of repulsive point process. DPP's provide exact methods to calculate the probability of a subset of items, that can be sampled, from a core set. DPP's are gaining increasing interest as they have proven effective in machine learning Kulesza et al. (2012) and multi-agent reinforcement learning Yang et al. (2020), having originated from modelling repulsive particles in quantum mechanicsMacchi (1977). It has been shown that extending the use of a determinantal point process (DPP) in multi-agent reinforcement Yang et al. (2020) learning can significantly improve joint exploration across agents.

**Definition 1 (DPP)** *For a ground set of items $\mathcal{Y} = \{1, 2, \ldots, M\}$, a DPP, denoted by $\mathbb{P}$, is a probability measure on the set of all subsets of $\mathcal{Y}$, i.e., $2^{\mathcal{Y}}$. Given an $M \times M$ positive semi-definite (PSD) kernel $\boldsymbol{\mathcal{K}}$ that measures similarity for any pairs of items in $\mathcal{Y}$, let $\boldsymbol{Y}$ be a random subset drawn according to $\mathbb{P}$, then we have, $\forall Y \subseteq \mathcal{Y}$,*

$$\mathbb{P}_{\boldsymbol{\mathcal{K}}}(\boldsymbol{Y} = Y) \propto \det(\boldsymbol{\mathcal{K}}_Y) = \mathrm{Vol}^2(\{\boldsymbol{w}_i\}_{i \in Y}), \tag{3}$$

*where $\boldsymbol{\mathcal{K}}_Y := [\boldsymbol{\mathcal{K}}_{i,j}]_{i,j \in Y}$ denotes the submatrix of $\boldsymbol{\mathcal{K}}$ whose entries are indexed by the items included in $Y$.*

Where the diagonal values $\boldsymbol{\mathcal{K}}_{i,i}$ captures the quality of item $i$, and the off-diagonal values $\boldsymbol{\mathcal{K}}_{i,j}$ measures the diversity between items $i$ and $j$ with respect to some diversity function. The normaliser can be computed as: $\sum_{Y \subseteq \mathcal{Y}} \det(\boldsymbol{\mathcal{K}}_Y) = \det(\boldsymbol{\mathcal{K}} + \boldsymbol{I})$, where $\boldsymbol{I}$ is an $M \times M$ identity matrix. The key to the success of DPP's is the ability to naturally trade-off between diversity and quality, which in reinforcement learning terms would enable it to trade-off between exploration-exploitation naturally.

# 3 Emergent Communication is Repulsive

In this section, we will detail how one could apply a RPP to Emergent Communication, explaining theoretically how the direction application of RPPs promotes efficient emergent communication.

### 3.0.1 Biased agents for speaking listening

First, we ask the question: why have a separate policy for the message and action policy, when in reality, actions and communication can be output jointly? e.g. a human typically takes actions that align with their verbal statements, and vice versa. For this reason, we create a new joint action-message space $\mathcal{U} = \mathcal{A} \times \mathcal{M}$ for agents which speak and act. where $\times$ is the Cartesian product. Of course, this new action space will have poor scalability; however, it will enable us to reason over actions and messages jointly.

Since the diagonal and off-diagonal entries of $\mathcal{K}$ represent *quality* and *diversity* respectively, we allow a decomposition of the similarity matrix similar to Yang et al. (2020) $\mathcal{K} := \mathcal{D}\mathcal{F}\mathcal{D}^T$ with $\mathcal{D} \in \mathbb{R}^{N \times N}$ and $\mathcal{F} \in \mathbb{R}^{N \times N}$ with each row $\mathbf{K}_i = d_i^2 \mathbf{f}_i^T$ is a product of a **quality** term $d_i^2 \in \mathbb{R}^+$ and a **diversity** feature vector $\mathbf{f}_i \in \mathbb{R}^{N \times 1}$. Where each (i-th) row of $\mathcal{F}$ is ensured to have unit norm by dividing through by $\|\mathbf{f}_i\|$.

We can compute the **diversity** matrix $\mathcal{F}$ through any similarity function, such as euclidean distance $\mathcal{F}_{i,j} = \|[\hat{\mathbf{o}}_i, \mathbf{u}_i] - [\hat{\mathbf{o}}_j, \mathbf{u}_j]\|^2$ between concatenated observation-message-action points $[\mathbf{o}_i, \mathbf{a}_i]$ and $[\mathbf{o}_j, \mathbf{a}_j]$.

If we denote the quality term for a given observation-action pair as $d_i := \exp\left(\frac{1}{2}Q(\hat{o}_i, u_i)\right)$ and setting $\mathcal{Y} = \left\{(\hat{o}_1^1, u_1^1), \ldots, (\hat{o}_N^{|\mathcal{O}| \times |\mathcal{M}|}, u_N^{|\mathcal{A}| \times |\mathcal{M}|})\right\}$, $\mathcal{C}(\mathbf{o}) := \left\{Y \subseteq \mathcal{Y} : |Y \cap \mathcal{Y}_i(o_i)| = 1, \forall i \in \{1, \ldots, N\}\right\}$, with $\mathcal{Y}_i(o_i)$ of size $|\mathcal{A}| \times |\mathcal{M}|$ and $|\mathcal{C}(\mathbf{o})| = (|\mathcal{A}| \times |\mathcal{M}|)^N$ we can then formulate our DPP distribution over a mini-batch of data by;

$$\tilde{\mathbb{P}}_{\mathcal{K}}\left(\mathbf{Y} = Y | \mathbf{Y} \in \mathcal{C}(\mathbf{o})\right) \propto \log\det\left(\mathcal{K}\right)$$

$$\propto \log\det\left(\mathcal{D}_Y \mathcal{F}_Y \mathcal{D}_Y^T\right)$$

$$\propto \log\left(\det\left(\mathcal{D}_Y^T\right)\det\left(\mathcal{F}_Y\right)\det\left(\mathcal{D}_Y\right)\right) \tag{4}$$

$$\propto \sum_{i=1}^B Q(\hat{o}_i, u_i) + \log\det\left(\mathcal{F}_Y\right).$$

Using $\det(\mathbf{A}\mathbf{B}) = \det(\mathbf{A})\det(\mathbf{B})$, for square matrices $\mathbf{A}\&\mathbf{B}$.

### 3.0.2 Speaker

If we have an agent who is just listens or speaks , we can similarly derive the DPP action-observation/ message-observation distribution. Setting $\mathcal{Y} = \left\{(o_1^1, m_1^1), \ldots, (o_N^{|\mathcal{O}|}, m_N^{|\mathcal{M}|})\right\}$, $\mathcal{C}(\mathbf{o}) := \left\{Y \subseteq \mathcal{Y} : |Y \cap \mathcal{Y}_i(o_i)| = 1, \forall i \in \{1, \ldots, N\}\right\}$, with $\mathcal{Y}_i(o_i)$ of size $|\mathcal{M}|$ and $|\mathcal{C}(\mathbf{o})| = |\mathcal{M}|^N$, we can now derive the probability density function for the speakers message-observation distribution;

$$\tilde{\mathbb{P}}_{\mathcal{K}}^{\mathbf{Speaker}}\left(\mathbf{Y} = Y | \mathbf{Y} \in \mathcal{C}(\mathbf{o})\right)$$

$$\propto \sum_{i=1}^B Q(o_i, m_i) + \log\det\left(\mathcal{F}_Y\right). \tag{5}$$

### 3.0.3 Listener

Similarly by setting $\hat{\mathbf{o}} = [\mathbf{o}, \mathbf{o}^m]$ and $\mathcal{Y} = \left\{(\hat{o}_1^1, a_1^1), \ldots, (\hat{o}_N^{|\mathcal{O}|}, a_N^{|\mathcal{A}|})\right\}$, $\mathcal{C}(\hat{\mathbf{o}}) := \left\{Y \subseteq \mathcal{Y} : |Y \cap \mathcal{Y}_i(\hat{o}_i)| = 1, \forall i \in \{1, \ldots, N\}\right\}$, with $\mathcal{Y}_i(\hat{o}_i)$ of size $|\mathcal{A}|$ and $|\mathcal{C}(\hat{\mathbf{o}})| = |\mathcal{A}|^N$, we can now derive the probability density function for the listeners action-observation distribution

$$\tilde{\mathbb{P}}_{\mathcal{K}}^{\textbf{Listener}}\big(\boldsymbol{Y} = Y | \boldsymbol{Y} \in \mathcal{C}(\boldsymbol{o})\big)$$

$$\propto \sum_{i=1}^{B} Q\big(\hat{o}_i, a_i\big) + \log \det \big(\boldsymbol{\mathcal{F}}_Y\big). \tag{6}$$

One can see, that when a speaker is sampling message-observation pairs from this DPP, they will be biased towards **positive speaking** as the DPP will be biased to sampling mixed messages concerning different states, as this diversity of state-message pairs yields a larger determinant, thus a larger probability of being sampled as seen in Equation **?**. As the message quality value increases, we expect to see the agent more like to sample this message rather than other more diverse messages.

When a listener is sampling actions from this DPP, the DPP will promote **positive listening** as it will grant higher probabilities to sets of actions which are diverse concerning differing incoming messages. Similarly, as the action value increases, the agent will be less drawn to diversifying its actions and taking the one with higher value.

The last remaining challenge is sampling from a defined partitioned DPP's, as we are constrained by the fact that each observation requires an action, rather than the typical DPP which is allowed to free sample items from the core set. However, there is a wealth of solutions to this such as sampling-by-projection as seen in Celis et al. (2018) and Chen et al. (2018). Additionally, due to similar partitioning of the DPP one can expect sampling to simplify down to a scheme similar to Yang et al. (2020), where each row is sampled sequentially. We leave this up to future work.

## 4 Conclusion & Future Work

We examined properties of successful emergent communication and explained how they in-fact encourage a repulsive point processes over the actions/ messages. We hope that these theoretical connections between emergent communication and RPPs provides justification and inspiration for future researchs.

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
