# OpenReview forum: "Why Emergent Communication is Repulsive"
_TMLR — Rejected by TMLR_

### Review · Reviewer_L75y · 2022-04-22

**Summary Of Contributions:**

The paper makes a connection between learning emergent communication protocols and determinantal point processes (DPPs), a specific type of repulsive point process (RPP). Specifically, the paper frames some of the challenges of learning a communication protocol as "trading off between diversity and quality" -- meaning, trading off between messages (and actions) that are diverse (and thus presumably help learning), and those that give high reward. The paper then brings in DPPs, which essentially define a probability distribution for sampling from a set of items, and have a built in mechanism for trading off diversity and quality. The paper then shows a way of sampling actions in emergent communication according to a DPP.


**Requested Changes:**

Biggest changes: improving the clarity of the paper, and answering the questions posed above.

Adding empirical evidence for why DPPs might be useful in emergent communication would be very nice, though isn't strictly necessary.

Small edits --
- p1: theoretically connections -> theoretical connections
- many references are given in the wrong format; eg Jaques et al (2019) instead of (Jaques et al., 2019) -- eg bottom of page 2
- p2: theoretically bias’s -> biases
- the section titles for section 2 and 3 are almost the same, I'd like them to be more descriptive
- p4: "we have an agent who is just listens" -> who just listens
- sec 3: change subsubsections to subsections
- p5: Equation ?
- p5: "on for future researchs" -> research

**Strengths And Weaknesses:**

Strengths
- As far as I know, this is the first paper to make a connection between emergent communication and RPPs, thus the paper is novel.
- The general problem of 'figure out what's going on in emergent communication' is valuable and worth studying.

Weaknesses
- My biggest concern is with the clarity of the paper. It took me a while to understand the point of the paper -- in particular, I don't think the framing of emergent communication as a tradeoff between quality and diversity was well explained, and this is a core motivation of the paper. I put several more concrete questions in the 'Questions' section below.

- There is very little discussion or insight as to how this would practically be implemented, and there are no empirical results. This isn't a dealbreaker per se -- it's okay to not have empirical results if you make interesting theoretical connections -- but one of the motivations for the paper is:
"There exist auxiliary losses that
pressure the speaking/ listening agent to alter its short-term behaviour in response to messages consistently [...] but bear with them the additional difficulties of tuning extremely sensitive auxiliary loss parameters, as is the case with most auxiliary loss functions. Thus, there is a need for more simplified emergent communication algorithms that achieve success in challenging language games with less sensitivity to hyperparameters"
and there is no discussion of sensitivity to hyperparameters in the rest of the paper.

- I feel like the connection between DPPs and emergent communication is made at a fairly high level, and think it would benefit from a more thorough exploration of what happens when emergent communication protocols are written as DPPs (eg answering some of the questions below).

Questions
" In this setup, one typically has at least two agents which are de-centralised, yet have a
communication channel between them that might be detrimental to the overall performance of both agents, i.e. one agent might be able to see but not move, and needs to guide another agent towards a goal."
Why is the communication channel detrimental to performance? I'm not sure what is meant here. Agents can just learn not to use the channel, and then their performance is the same as without the channel. Based on the second part of this sentence, maybe what is meant is that the communication channel can help agents coordinate to solve tasks when they have different abilities?

"The properties of successful emergent communication [...] typically involve a trade-off between ensuring diversity of message-action pairs, conditioned on observations, and consistency, when the reward acquired, is considerable."
- I don't understand the second part of this sentence: "consistency, when the reward acquired, is considerable". Is the point that there is a trade-off between diversity and consistency? This is a central part of the motivation of the paper, and I'd like it to be explained more clearly.

- Why is the quality term for a given observation-action pair d_i defined in this way? What is Q? I assume this is the Q function, but it is not defined elsewhere in the paper. Does this mean that algorithms using DPPs need to learn a Q function?

- Why is the diversity matrix F calculated between concatenated observation-message-action pairs? This could use a bit more elaboration.

"For this reason, we create a new joint action-message space U = A × M
for agents which speak and act. where × is the Cartesian product. Of course, this new action space will
have poor scalability; however, it will enable us to reason over actions and messages jointly."
- Does this mean the DPP approach is only practical for small-scale emergent communication experiments?

- Given that in emergent communication, actions and messages are usually represented by one-hot vectors, it seems like the diversity matrix F is completely uninteresting (and so you are just sampling the highest value action / message that hasn't yet been sampled). Is this accurate? Or are you proposing learning some kind of message / action embeddings?

- It's not immediately clear to me how the last lines in (4), (5), and (6) would be used in a practical emergent communication algorithm -- a bit more elaboration here would be useful.

- What's the difference between "trading off between quality and diversity" and simply the exploration-exploitation tradeoff that is ubiquitous in RL (and emergent communication)?

---

> ### Author Response · Authors · 2022-05-14
> **Author Response**
>
> Dear reviewer,
>
> We appreciate the time taken. I As previously stated the intention of the paper is merely to show theoretical connections between the two fields, and I am happy that you appreciate this contribution.
>
> I will make all the necessary changes required and re-write the paper to improve clarity.
>
> "Is the point that there is a trade-off between diversity and consistency?" --> My use of consistency is indeed vague here, we refer to consistency as the ability to take high-quality decisions when they are near-optimal.
>
> "Why is the quality term for a given observation-action pair d_i defined in this way? What is Q? I assume this is the Q function, but it is not defined elsewhere in the paper. Does this mean that algorithms using DPPs need to learn a Q function?" --> Apologies, I previously tried to define a q learning algorithm and left some parts in there incorrectly labelled.
>
> "What's the difference between "trading off between quality and diversity" and simply the exploration-exploitation tradeoff that is ubiquitous in RL (and emergent communication)?" --> there is no difference, they are indeed the same meaning!
>
> "Given that in emergent communication, actions and messages are usually represented by one-hot vectors, it seems like the diversity matrix F is completely uninteresting (and so you are just sampling the highest value action/message that hasn't yet been sampled). Is this accurate? Or are you proposing learning some kind of message/action embeddings?" --> Exactly, I didn't make this clear either but following a similar formulation as done by Yang et al, who introduce learnt embeddings for objects involved in the diversity matrix.
>
> "concatenated observation-message-action diversity matrix" --> I will explain this more in the paper, but the idea is that when calculating these jointly you are able to model the diversity of responses to incoming messages/ actions. In EC its important to explore this joint response/ incoming information space efficiently, as this is exactly what these positive listening and signalling losses do, promote diverse responses from incoming messages during training.
>
> "DPP practicalities" -> Depends what we mean by small scale here, but yes due to the computation of DPP (which can be approximated with a linear time algorithm) it can cause scaling issues.
>
> Thanks!

---

### Review · Reviewer_YVNW · 2022-04-25

**Summary Of Contributions:**

The authors discuss previous work on emergent communication and bring up that positive listening and positive signaling has been useful properties to induce the emergence of such. Then they show that Determinant Point Processes (DPPs) have these properties.


**Broader Impact Concerns:**

No concerns.

**Requested Changes:**

Please perform an experimental evaluation, at least of an illustrative "proof of concept" form.

**Strengths And Weaknesses:**

The result is only an indication that Determinant Point Processes could be of interest for emergent communication, while an experimental section investigating how this works out is missing. They point out that DPPs are repulsive, and that this is causing different actions for different incoming observations to be different. This would argue that repulsivity might help with having emergent communication, but perhaps does not really say that emergent communication can only happen during such constraints (as the title might be read as implying) nor that will if they are satisfied.

In the final paragraph the authors discuss how one could actually practically use DPPs for experiments on emergent communication, but say they leave that to future work. However, I am of the view that carrying out that part  could help answer how useful DPPs are for emergent communication and that the article cannot conclude that much without such.

---

> ### Author Response · Authors · 2022-05-14
> **Experimental evaluation**
>
> Dear reviewer,
>
> Apologies for the slow reply, we thank you for your time and comments.
>
> Please see the response to Reviewer t857 regarding removing proposed losses.
>
> Thanks!

---

### Review · Reviewer_t857 · 2022-04-28

**Summary Of Contributions:**

The paper proposes using repulsive point processes, which are probabilistic models that allow sampling diverse and qualitative samples within a group of finite sets, to provide a theoretically motivated objective function to train emergent languages.

**Broader Impact Concerns:**

No broader impact concerns.

**Requested Changes:**

As highlighted in the abstract, the main advantage of using DPP over other frameworks is to yield efficient emergent communication but there is no evidence of how it performs in practice. Almost every work in EC is evaluated on various benchmarks, it is not possible to evaluate this work with any prior research without any empirical evidence.

I believe this manuscript is an interesting work in progress that could lead to novel insights in emergent communication training with an experimental section.

**Strengths And Weaknesses:**

The paper makes a connection between the proposed repulsive point processes and the two key aspects of emergent communication training i.e. Positive Listening and Positive Signalling. The idea is to promote diversity among the messages while also creating consistency between the message-observation pair.

Although the motivation behind developing probability density functions using DPP is interesting, it is still not clear how it changes the objective function for both speaker and listener such that it would lead to efficient emergent communication. In both pdfs, the resulting distribution seems to be a linear sum of a diversity term and a quality term. Given that the authors already cite the relevant works (Lowe et al. 2019, Eccles et al. 2019a) that use this approach, it is not clear how having access to these density functions would lead to a different and more stable training manifold.

---

> ### Author Response · Authors · 2022-05-14
> **Response**
>
> Dear reviewer,
>
> Apologies for the slow reply, we thank you for your time and comments.
>
> The purpose of this paper was merely a theoretical connection between the two fields rather than an empirical paper. As highlighted by Reviewer L75y, this is the first theoretical justification for tricks used in EM that drastically improve results, rather than intuition provided by the authors of the auxiliary loss functions. We believe justification and explanation to be of equal importance to empirical results, as it aids the interpretability of the overall system.
>
> If I were to remove the proposed new losses would this then be satisfactory?
>
> Thanks!

---

### Decision · Action_Editors · 2022-06-11

**Recommendation:** Reject

**Comment:**

All reviewers agreed that the theoretical connections proposed are interesting, but in their current state, not clear enough to warrant acceptance without supporting experiments. I encourage the authors to resubmit with reviewer questions addressed and, ideally, supporting experiments, even if illustrative.